# Multipocket synergy towards high thermoelectric performance in topological semimetal TaAs$_2$

Haihua Hu [1,3], Xiaolong Feng [1,3], Yu Pan [2] ✉, Vicky Hasse[1], Honghui Wang[1], Bin He [1] ✉ & Claudia Felser [1] ✉

Charge-carrier compensation in topological semimetals amplifies the Nernst signal and simultaneously degrades the Seebeck coefficient. In this study, we report the simultaneous achievement of both a large Nernst signal and an unsaturating magneto-Seebeck coefficient in a topological nodal-line semimetal TaAs$_2$ single crystal. The unique dual-high transverse and longitudinal thermopowers are attributed to multipocket synergy effects: the combination of a strong phonon-drag effect and the two overlapping highly dispersive conduction and valence bands with electron–hole compensation and high mobility, promising a large Nernst effect; the third Dirac band causes a large magneto-Seebeck effect. High transverse and longitudinal power factors of ~3100 and ~50 μW cm$^{-1}$ K$^{-2}$, respectively, are achieved, surpassing those of other topological semimetals and mainstream semiconductors. Our study presents a feasible approach for optimizing the longitudinal and transverse thermopowers in topological semimetals simultaneously and demonstrates the potential of TaAs$_2$ for low temperature solid-state cooling.

Explorations of the intriguing transport properties and quantum behaviors of topological semimetals have enriched the scope of topological quantum materials for advanced energy conversion technologies, such as spin-caloritronics, topological catalysis, and thermoelectrics[1–5]. Thermoelectric technology, which realizes the direct conversion between heat and electricity, has significant potential for power generation from waste heat and electronic refrigeration via solid-state cooling[6–8]. The output power of a thermoelectric device is determined by the power factor[9], defined as $PF = S^2/\rho$, where $S$ and $\rho$ represent the thermopower and resistivity, respectively. Depending on the configuration of the output voltage and applied temperature gradient, there exists a longitudinal Seebeck effect and a transverse Nernst effect, where the output voltage is along the same direction as the temperature gradient in the Seebeck effect, whereas the output voltage is perpendicular to the temperature gradient in the Nernst effect under magnetic fields. Therefore, the Seebeck and Nernst power

factors are defined as $PF_S = S_{xx}^2/\rho_{xx}$ and $PF_N = S_{yx}^2/\rho_{yy}$, respectively, where $S_{xx}$ and $S_{yx}$ represent Seebeck and Nernst thermopower, respectively. In recent years, topological semimetals have attracted significant attention for the Nernst effect because of their exceptionally high performance[10–15]. Further, the Nernst device fabrication process is significantly simplified because the indispensable paired assemblies encountered in the Seebeck device are unnecessary, which helps to reduce the electrical and thermal resistance in the modules[16,17].

A more efficient and straightforward cooling system capable of multi-directional refrigeration can be achieved by combining transverse Nernst and longitudinal Seebeck thermopowers[18]. A high $S_{yx}$ is expected in a two-carrier system with high mobility and electron–hole compensation because electrons and holes are deflected in opposite directions by the magnetic field in the Nernst effect. However, the two-carrier transport behavior is detrimental to the Seebeck coefficient because the joint transport of electrons and holes in the same

[1]Max Planck Institute for Chemical Physics of Solids, Nöthnitzer Str. 40, Dresden, Germany. [2]College of Materials Science and Engineering and Center of Quantum Materials & Devices, Chongqing University, Chongqing 400044, China. [3]These authors contributed equally: Haihua Hu, Xiaolong Feng.
✉e-mail: yupan2024@cqu.edu.cn; Bin.He@cpfs.mpg.de; Claudia.Felser@cpfs.mpg.de

direction reduces the accumulated Seebeck voltage. Consequently, it is considerably challenging to achieve high Nernst thermopower and Seebeck coefficients simultaneously in a single material. Solving this dilemma relies on introducing electrons and holes with different mobilities in different pockets. For example, in Dirac or Weyl semimetals with high mobilities, a large magneto-Seebeck effect can be experimentally expected when the topological quantum effect is introduced because of the linear band dispersion $E = \hbar k v$[19,20]. Meanwhile, electrons and holes in other pockets that have lower mobilities can contribute to the Nernst effect.

In this work, we simultaneously achieved large longitudinal and transverse thermopowers in a nodal-line topological semimetal $TaAs_2$ single crystal using a multipocket synergy strategy (Fig. 1a). Since the topological semimetal $TaAs_2$ has a centrosymmetric monoclinic structure with a space group of $C2/m$ (no. 12), the Berry curvature is zero and the anomalous Nernst effect is not considered, even under magnetic since the external magnetic field is insufficient to induce a decent non-zero Berry curvature in $TaAs_2$, which is different from $Cd_3As_2$ observed by previous studies[21,22]. Theoretical calculations were conducted to investigate the effect of multipocket synergy on the

dual-high thermoelectric thermopowers of $TaAs_2$. The electronic band structures of $TaAs_2$ without and with spin-orbit coupling (SOC) are shown in Fig. 1b and Supplementary Fig. 1. Without the SOC, the anticrossing band near points A and M formed nodal lines. These nodal lines can be classified into two types within the Brillouin zones: one type includes two open spiral nodal lines that extend across the Brillouin zones through point A, while the second type comprises two closed nodal loops near the M point (Fig. 1c). After accounting for the SOC effect, the nodal lines gap out the anticrossing feature[23,24], resulting in a massive Dirac fermion that aligns with the zero-field massive Dirac dispersion relation[19]

$$E_{Dirac} = \sqrt{\left(\frac{\Delta}{2}\right)^2 + \hbar^2 v_F^2 K^2} \tag{1}$$

where $\Delta$ and $v_F$ represent the energy gap and Fermi velocity, respectively (Supplementary Fig. 1c–e). Therefore, there are three Fermi pockets crossing the Fermi level $E_F$, where the hole pocket centered at the M point and electron pocket close to the M point can result in an electron–hole compensation behavior, and the massive Dirac pockets

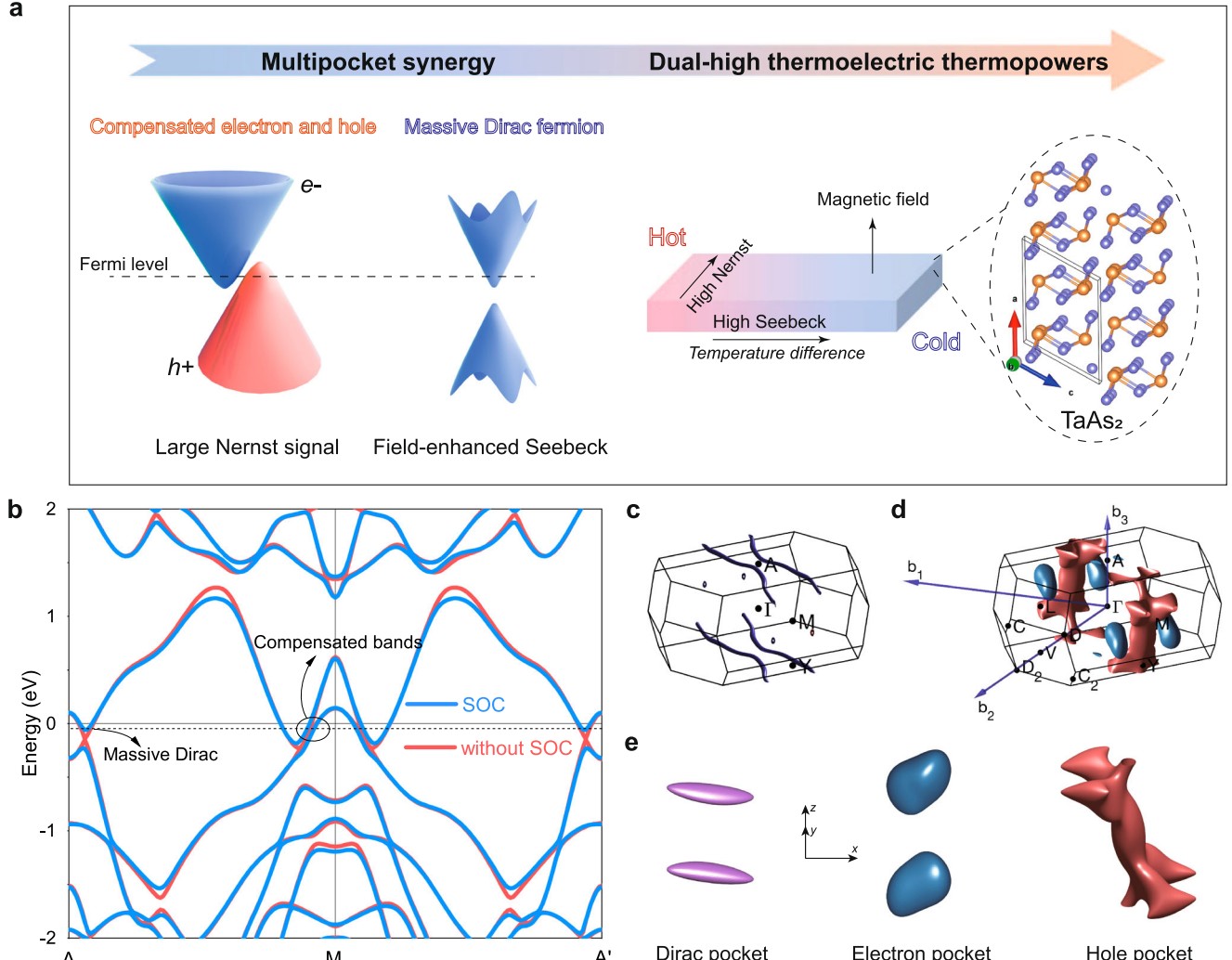

**Fig. 1 | Multipocket synergy leading to high thermopowers. a** Schematic of multipocket synergy in the $TaAs_2$ crystal that boost the high-thermoelectric thermopower. The pockets with electron–hole compensation contribute to a large Nernst signal, and a massive Dirac pocket enhances the Seebeck coefficient under an applied field. **b** Electronic band structures of $TaAs_2$ with and without spin-orbit coupling. Band crossings along the M-A direction are presented without the SOC. The dashed line represents the actual Fermi level. **c** Calculated nodal lines in the first Brillouin zone. **d, e** Fermi surfaces of two bands with spin-orbit coupling calculations. Fermi pockets in (**e**) correspond to the massive Dirac pocket (purple), compensated electron pocket (blue), and compensated hole pocket (red).

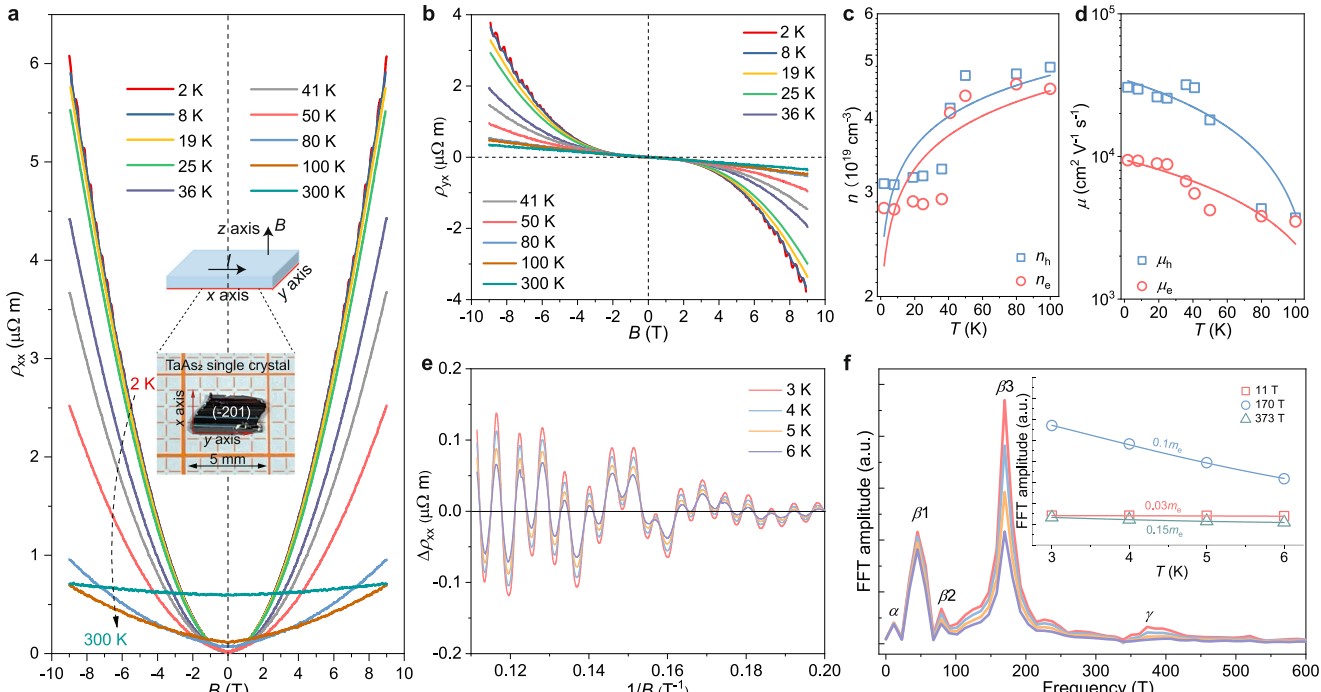

**Fig. 2 | Electrical transport properties and quantum oscillations. a** Magnetic field dependence of longitudinal resistivity $\rho_{xx}$. The inset displays a photograph of the TaAs$_2$ single crystal shaped like a bar with a length of ~5 mm and a width of 3 mm. The ($\bar{2}$01) crystal plane is the largest exposed surface of the TaAs$_2$ single crystal. The schematic indicates the electrical current along the $x$ axis and magnetic field along the $z$ axis. **b** Hall resistivity $\rho_{yx}$ as a function of the magnetic field. **c** Carrier concentration. **d** Carrier mobility. **e** Amplitudes of resistivity oscillations

as a function of $1/B$ obtained by subtracting a continuous polynomial. **f** Fast Fourier transform (FFT) spectra at different temperatures showing peaks corresponding to holes and electron pockets. The frequencies $\alpha$, $\beta$, and $\gamma$ represent the massive Dirac pocket, compensated hole pocket, and compensated electron pocket, respectively. The inset displays the effective mass calculations of different Fermi pockets from the Lifshitz–Kosevich formula.

around point A may produce a large magneto-Seebeck effect because of the quantum effect[19,20]. Figure 1d shows the three types of Fermi surfaces present in the first Brillouin zone of TaAs$_2$. The hole pocket is located at the M point, while the large electron pocket can be found close to the M point, and there is also a small electron pocket near point A. The mapped Fermi surfaces (Fig. 1e) reveal that the electron (blue) and hole (red) pockets possess similar volumes, indicating a nearly perfect compensation of electrons and holes near the Fermi level.

## Results

### Electrical transport properties

High-quality TaAs$_2$ single crystals (Supplementary Figs. 2, 3, and Fig. 2a) were used to investigate the transport properties. The TaAs$_2$ single crystal exhibited a distinguishable positive magnetoresistance (Fig. 2a, Supplementary Fig. 4) without reaching saturation up to 9 T at low temperatures, suggesting typical semimetal behavior. The distinct nonlinear field dependence of the Hall resistivity (Fig. 2b) indicates the collaborative contribution of both hole and electron carriers to electrical transport, which is consistent with the band structure calculations. Furthermore, carrier concentration and mobility were determined using a semiclassical two-carrier model based on longitudinal and Hall resistivity measurements[25]. The fitting formulae and results are detailed in Supplementary Fig. 5, and the carrier parameters are presented in Fig. 2c, d. The concentrations of electrons ($n_e$) and holes ($n_h$) are comparable over the entire temperature range, indicating a near compensation of holes and electrons. For example, $n_e$ and $n_h$ are ~$2.8 \times 10^{19}$ cm$^{-3}$ and ~$3.1 \times 10^{19}$ cm$^{-3}$ at 2 K, respectively. When the temperature ranges from 2 to 100 K, $n_e$ and $n_h$ exhibit a similar increase. The compensation of charge carriers and relatively low carrier concentrations in the TaAs$_2$ crystals provide additional evidence

for their semi-metal nature. Besides the carrier compensation, high charge carrier mobilities also play a critical role in determining the large Nernst thermopower[12]. The hole mobility ($\mu_h$) of the TaAs$_2$ crystal is ~$3.1 \times 10^4$ cm$^2$ V$^{-1}$ s$^{-1}$ at 2 K, whereas the electron mobility ($\mu_e$) is ~$0.94 \times 10^4$ cm$^2$ V$^{-1}$ s$^{-1}$. These values validate the superior quality of the TaAs$_2$ crystal and can be compared to those of topological semimetals that have been previously reported, such as NbSb$_2$ ($\mu_e = 2.1 \times 10^4$ cm$^2$ V$^{-1}$ s$^{-1}$ and $\mu_h = 1.2 \times 10^4$ cm$^2$ V$^{-1}$ s$^{-1}$ at 5 K)[10], and NbAs$_2$ ($\mu_e = 3.6 \times 10^4$ cm$^2$ V$^{-1}$ s$^{-1}$ and $\mu_h = 7.2 \times 10^4$ cm$^2$ V$^{-1}$ s$^{-1}$ at 2 K)[13].

Further, both $\rho_{xx}$ and $\rho_{yx}$ exhibit distinct quantum Shubnikov–de Haas (SdH) oscillations at magnetic fields above 5 T below 8 K, helping analyze the shape of Fermi surfaces[26–28]. The variation trend of the SdH oscillation amplitudes with an inverse magnetic field is shown in Fig. 2e, demonstrating their sensitivity to temperature differences. The amplitude of the longitudinal resistivity oscillations progressively weakened with increasing temperature. A fast Fourier transform (FFT) analysis of the oscillatory part revealed that there are five fundamental frequencies at $F = 11$ T ($\alpha$), 45 T ($\beta$1), 79 T ($\beta$2), 179 T ($\beta$3), and 373 T ($\gamma$) (Fig. 2f), which are in good agreement with those reported in the previous works[29,30]. Frequencies $\alpha$, $\beta$, and $\gamma$ are assigned to the massive Dirac pocket, compensated hole pocket, and compensated electron pocket, respectively. We can determine the effective masses $m^*$ (inset of Fig. 2f) of the three Fermi pockets using the Lifshitz–Kosevich formula by analyzing the temperature dependence of the SdH amplitudes $A_{FFT}$: $A_{FFT} = A_0 G / \sinh(G)$, where $A_0$ represents a constant, $G = 14.69 \, m^* T / B$, $1/B = 1/B_1 + 1/B_2$, and $B_1$ and $B_2$ represent the initial and final magnetic fields of the FFT field window[26]. According to the fitting results, the effective masses of the three Fermi pockets are $0.03 m_e$ ($\alpha$), $0.1 m_e$ ($\beta$3), and $0.15 m_e$ ($\gamma$), with $m_e$ representing the free electron mass. Therefore, the high carrier mobility of the TaAs$_2$ crystal can be attributed to its small effective mass and linear dispersion bands.

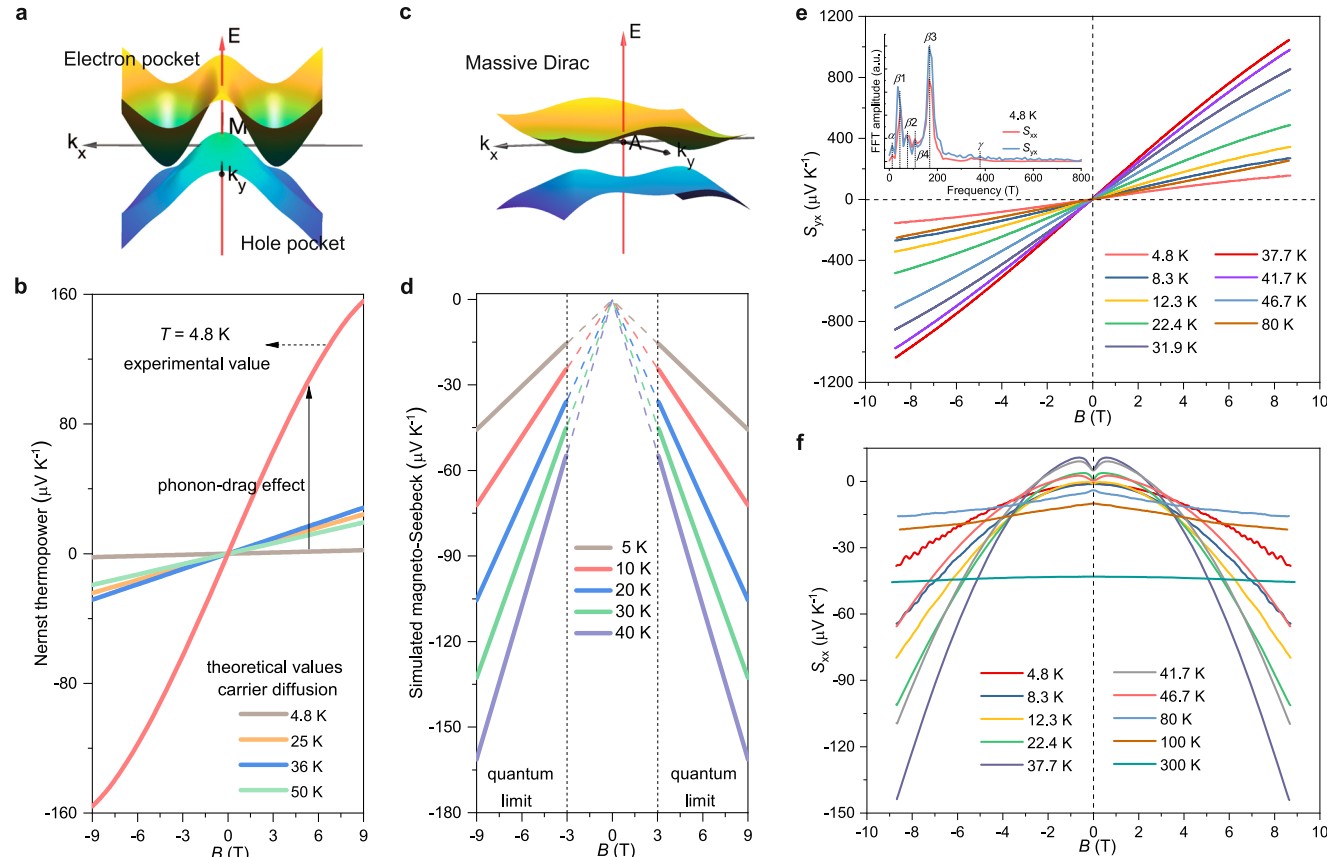

**Fig. 3 | Thermoelectric thermopowers. a** Schematic of the compensated electron–hole pockets near the Fermi energy. **b** Field-dependent Nernst thermo-power related to the charge carrier diffusion processes (calculation) and phonon-drag effect (experiment). **c** Spin-orbit coupling gapped the nodal lines into massive Dirac pockets close to point A. **d** The magnetic field-dependent Seebeck coefficient of the massive Dirac pocket in TaAs$_2$ based on the theoretical model in the extreme quantum limit ($\pm 3 \sim \pm 9$ T). Magnetic field dependence of (**e**) Nernst thermopower $S_{yx}$, and **f** Seebeck coefficient $S_{xx}$ at different temperatures. The inset shows the FFT spectra of $S_{xx}$ and $S_{yx}$ at 4.8 K.

## Thermal transport properties

Compensated high mobility electrons and holes near the Fermi level, along with a strong phonon-drag effect, can lead to exceptional transverse thermoelectric performance. Figure 3a shows the compensated electron–hole pockets we focused on for calculating the Nernst thermopower. The Nernst thermopower related to the charge carriers diffusion processes at different temperatures was calculated based on the experimental mobilities, carrier concentrations, and effective masses (Fig. 3b). Further information on how this calculation was carried out is provided in Supplementary Note 1. The experimental Nernst thermopower reveal a maximum value of 156 μV K$^{-1}$ at 4.8 K and 9 T, significantly higher than the thermopower related to the charge carriers diffusion processes (-2 μV K$^{-1}$) at the same temperature and magnetic field. The exceptional large Nernst thermopower at low temperatures is usually attributed to the phonon-drag effect[10]. The experimental Nernst thermopower $S_{yx}$ of the TaAs$_2$ single crystal as a function of the magnetic field is shown in Fig. 3e. $S_{yx}$ is nonlinearly corrected with unsaturated values with a magnetic field ranging from 4.8 to 46.7 K; however, it starts to deviate from this pattern gradually once the temperature exceeds 46.7 K. The transition from a nonlinear to linear Nernst signal in TaAs$_2$ single crystal indicates a slight discrepancy in carrier compensation at temperatures below 46.7 K, while an almost perfect compensation can be obtained once the tempera-ture surpasses this threshold. With increasing temperature, the $S_{yx}$ value rises under a magnetic field of 9 T, reaching a peak of -1045 μV K$^{-1}$ at 37.7 K before decreasing at higher temperatures. Moreover, a two-carrier model was used to analyze the behavior of the Nernst ther-mopower in TaAs$_2$. The Seebeck coefficients for electrons ($S_{xx}^e$) and

holes ($S_{xx}^h$) deviate from linear temperature dependence below 100 K and exhibit peaks around 45 K, indicating a significant contribution from phonons ($S_p$) to the Seebeck coefficient at low temperatures. The absolute values of $S_p^e$ and $S_p^h$ reach maximum values of 73 μV K$^{-1}$ and 76 μV K$^{-1}$ around 45 K, which are much larger than the values of $S_d^e$ (4.1 μV K$^{-1}$) and $S_d^h$ (2.6 μV K$^{-1}$) at the same temperature. This suggests that the phonon-drag effect significantly enhances the total Nernst effect in single-crystalline TaAs$_2$[10,31]. Additional details can be found in Supplementary Note 1 and Supplementary Fig. 6. Additionally, the thermal conductivity behavior (Supplementary Fig. 7) can be used as further evidence to support the phonon-drag effect.

Unlike the Nernst signal, understanding the magneto-Seebeck effect of a massive Dirac pocket (Fig. 3c) requires the involvement of a topological quantum picture. The thermopower of a Dirac semimetal undergoes quantum oscillations when an increasing magnetic field is applied because of the depopulation of higher Landau levels. Once the magnetic field reaches a certain strength where $\hbar v_F/l_B > E_F$ ($l_B = \sqrt{\hbar/eB}$ represents the magnetic length), the system crosses into the extreme quantum limit, resulting in a notable magnetic field-induced variation in the Fermi energy ($\mu \propto 1/B$) and density of states ($D(\mu) \propto B$)[20]. In the extreme quantum limit, the Seebeck coefficient can be expressed as $S_{xx} \simeq \frac{\kappa_B^2 NTB}{6\hbar^2 v_F C}$, where $\kappa_B$, $N$, $T$, and $C$ represent the Boltzmann constant, number of Dirac nodes, temperature, and carrier concentration, respectively. As the density of states increases, $S_{xx}$ also exhibits a linear non-saturated increase with the magnetic field, leading to a large Seebeck coefficient. More quantitatively, the theoretical simulation results in Fig. 3d show a linear correlation between the Seebeck

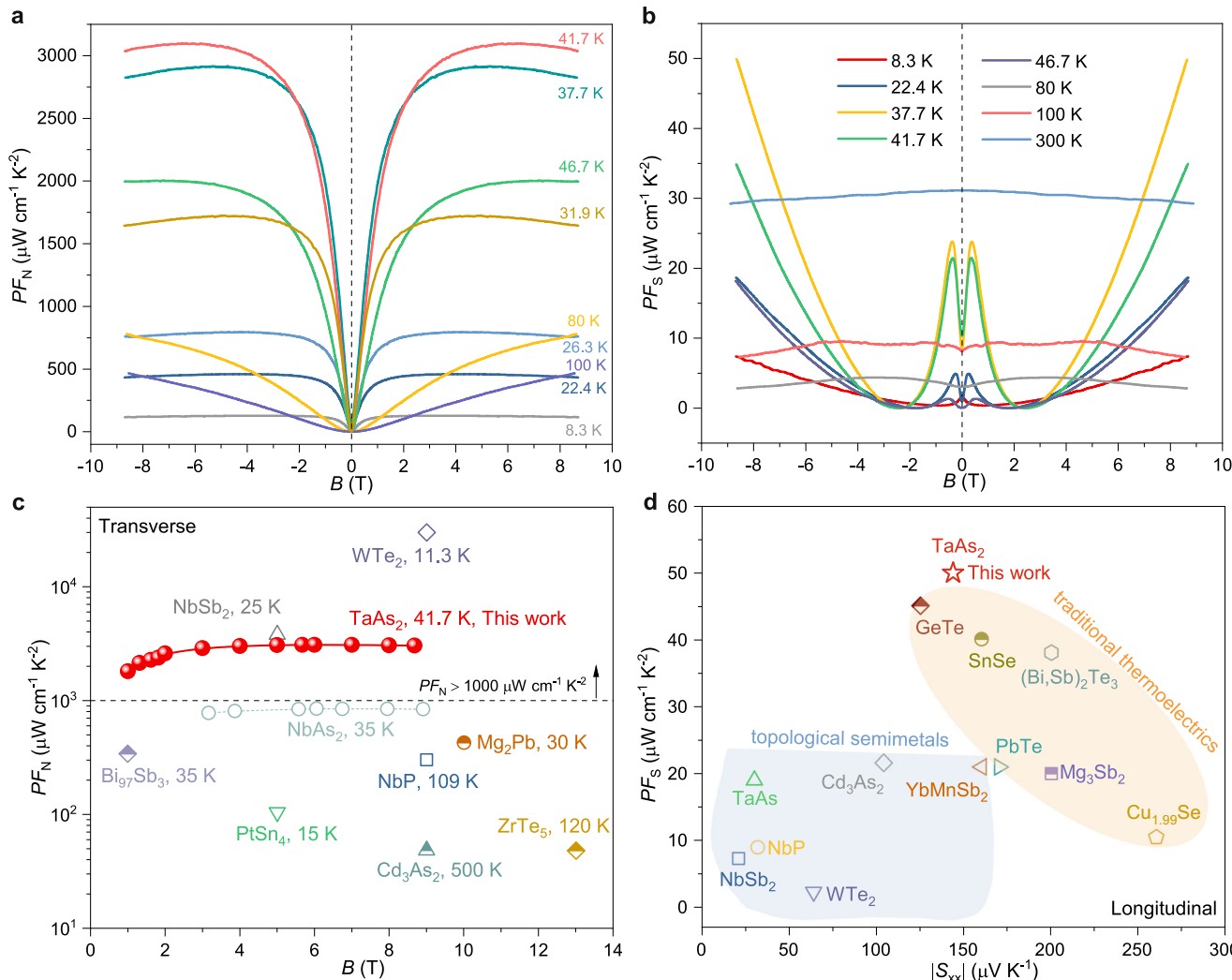

**Fig. 4 | Power factors.** Magnetic field dependence of (**a**) transverse power factor $PF_N$ and (**b**) longitudinal power factor $PF_S$ at different temperatures. Comparison of the (**c**) Nernst power factor and **d** longitudinal power factor of TaAs$_2$ with those of reported topological semimetals and thermoelectric semiconductors[10–13,18,35–37,41,48–56]. In order to eliminate the influence of other mechanisms, all the data are taken from single crystals and polycrystals without any further doping or alloying.

coefficient of TaAs$_2$ and the magnetic field, confirming that the large Seebeck coefficient originates from a massive Dirac fermion.

We measured the magnetic-field-dependent Seebeck coefficient $S_{xx}$ to investigate the influence of the massive Dirac fermion on the longitudinal thermoelectric performance, as depicted in Fig. 3f. The reversal of the $S_{xx}$ sign at various temperatures or magnetic fields is linked to the transition of the carrier type between electrons and holes, potentially caused by the slight shift in the Fermi level, which results in asymmetric conduction and valence bands[32,33]. A remarkably low $S_{xx}$ value of less than 15 µV K$^{-1}$ in the absolute value is obtained when the temperature is below 100 K and the magnetic field is less than 2.5 T. With the magnetic field exceeding 2.5 T, the $S_{xx}$ data demonstrates a more noticeable upward trend as the magnetic fields intensify across the temperature range, particularly exhibiting an unsaturated super linear increase between 22.4 K and 41.7 K. Under 9 T and 37.7 K, the maximum $S_{xx}$ reaches about −144 µV K$^{-1}$, -27 times the value at 0 T. The unsaturated magneto-Seebeck effect is caused by the massive Dirac pocket entering the extreme quantum limit, resulting a linear increase. Moreover, evident quantum oscillations are detected in both the Nernst thermopower and Seebeck coefficient below 8.3 K. The FFT analysis of the $S_{xx}$ and $S_{yx}$ oscillations at 4.8 K reveals five main frequencies, which is in accordance with the magnetoresistance results,

as depicted in the inset of Fig. 3e and Supplementary Fig. 6. Moreover, an additional frequency $\beta 4$ (106 T) is observed, possibly related to an irregular hole pocket as reported previously. The thermal transport oscillation, being the energy derivative, is more sensitive than the SdH, explaining why this frequency is not observed in the SdH analysis[29,34].

**Thermoelectric power factors**

In addition to the large Nernst thermopower and Seebeck coefficient, the transverse and longitudinal power factors are key for determining the heat-pumping power and output power density of thermoelectric devices. Figure 4a, b show the magnetic-field-dependent transverse and longitudinal power factors, respectively. At temperatures below 80 K, the $PF_N$ showed a rapid increase in low fields, eventually leveling off at a saturated value. This results in a peak value of ~3100 µW cm$^{-1}$ K$^{-2}$ at 9 T and 41.7 K, ranking at the top level among topological materials (Fig. 4c). An exceptional high $PF_N$ (exceeding 460 µW cm$^{-1}$ K$^{-2}$) can be retained even at a high temperature of 100 K. When the temperature falls below 46.7 K and the magnetic field goes above 2.5 T, the $PF_S$ of the TaAs$_2$ crystal exhibits a sharp rise, peaking at 50 µW cm$^{-1}$ K$^{-2}$ at 37.7 K and 9 T. Even at 300 K, the $PF_S$ remains at 31 µW cm$^{-1}$ K$^{-2}$ at 0 T, comparable to that of the state-of-the-art thermoelectric materials such as PbTe (-21 µW cm$^{-1}$ K$^{-2}$)[35], Mg$_3$Sb$_2$ (-20 µW cm$^{-1}$ K$^{-2}$)[36], and

$Cu_{1.99}Se$ (-10.5 μW cm$^{-1}$ K$^{-2}$)[37] (Fig. 4d). Moreover, similar to the case for most topological semimetal single crystals, $TaAs_2$ demonstrates high thermal conductivity $\kappa$ (see Supplementary Fig. 7). Considering the large power factors and high thermal conductivity, we can obtain transverse ($Z_N$, $Z_NT$, defined as $Z_N = PF_N/\kappa$) and longitudinal figure of merits ($Z_S$, $Z_ST$, defined as $Z_S = PF_S/\kappa$, Supplementary Fig. 8); their variation trend is similar to that of $PF_N$ and $PF_S$. A potential reduction in thermal conductivity can be achieved by reducing the carrier concentration and enhancing phonon scattering in future studies to achieve the high $Z_NT$[38–40]. Thus, the as-prepared $TaAs_2$ single crystal is a promising candidate for thermoelectric cooling because it combines longitudinal and transverse thermoelectric effects.

## Discussion

The coexistence of dual-high magneto-Seebeck coefficients and Nernst thermopowers achieved through a multipocket synergy strategy has rarely been reported. Prior to this study, Fu et al. had already investigated the Nernst power factor in polycrystalline NbP through a multipocket strategy[41]. Nevertheless, the motivation and outcomes of our study diverge significantly from theirs. While they relied on a combination of small and large electron pockets to compensate for a large hole pocket for enhanced Nernst thermopowers and power factors, they did not delve into the significance of the small pocket in the Seebeck coefficient. Additionally, it should be noted that $ZrTe_5$ is the only material known to possess large values of both magneto-Seebeck coefficients and Nernst thermopowers concurrently. Unlike $ZrTe_5$, which is recognized as a single-carrier system, the $TaAs_2$ crystal investigated in this study is a multipocket system with a massive Dirac pocket that contributes to its remarkable non-saturating Seebeck coefficient. The absence of intervalley scattering, which normally balances the carrier concentrations between different pockets and decreases both the Seebeck and Nernst thermopowers, can be attributed to the low temperature, where the phonons responsible for scattering are not yet active, preventing interactions between the electrons in each pocket. However, as the temperature increased, the phonon wave vector also increased. When the phonon wave vector approached the size of the Brillouin zone, strong scattering occurred, causing the two pockets to merge and decrease both $S_{xx}$ and $S_{yx}$. In addition, we discovered that the thermal conductivity reached its peak at 37.7 K, which was similar to the peaks of $S_{xx}$ and $S_{yx}$. Below 40 K, the mobilities of both electrons and holes remained relatively constant; however, above this temperature, the mobility decreased rapidly, leading to a decrease in $S_{yx}$.

The multipocket synergy strategy was successfully employed to optimize both the longitudinal and transverse thermopowers in the topological semimetal $TaAs_2$ single crystal, yielding a high Nernst power factor of approximately 3100 μW cm$^{-1}$ K$^{-2}$ and a high Seebeck power factor of ~50 μW cm$^{-1}$ K$^{-2}$. This strategy provides opportunities for improving the thermoelectric performance of topological semimetals, which represents a significant advancement in thermoelectric performance modulation. Hence, other two-carrier topological semimetals, such as $NbSb_2$[10], $NbAs_2$[13], $VP_2$, and $VAs_2$ (Supplementary Fig. 9), which possess two compensated pockets and an extra massive Dirac pocket, could serve as promising materials for achieving simultaneous enhancement of both the transverse and longitudinal thermoelectric performances by adjusting the Fermi level. Furthermore, topological semimetals exhibit considerable potential in applications relevant to longitudinal Peltier cooling and transverse Ettingshausen cooling because of the multipocket synergy effect. This can result in a more efficient and straightforward thermoelectric cooling system capable of multidirectional refrigeration.

## Methods
### Sample synthesis and characterization
$TaAs_2$ single crystals were synthesized using the chemical vapor transport method. Initially, a blend of high-purity tantalum and arsenic powders in a 1:2 molar ratio, were mixed with 0.1 g of iodine as the transport agent. The mixture was subsequently placed in an evacuated fused silica ampoule and allowed to react at 1073 K for more than 90 h. Subsequently, the transport reaction was conducted in a two-zone furnace with a temperature gradient ranging from 1273 to 1173 K over several weeks. Upon the completion of the reaction, the ampoule was removed from the furnace and rapidly quenched in water. Finally, bar-shaped crystals with shiny surfaces are obtained. Single crystallinity was examined using Laue X-ray diffraction, and the chemical composition was analyzed using scanning electron microscopy (Philips XL30) and Oxford energy-dispersive X-ray spectroscopy (Quantax, Bruker).

### Transport property measurements
The longitudinal and Hall resistivities were measured using a physical property measurement system (PPMS9, Quantum Design) in an electrical transport option with a standard heater and two thermometers. The Nernst thermopower, Seebeck coefficient, and thermal conductivity were also measured using the PPMS9 under high vacuum via a standard four-contact steady-state method. To calibrate any contact misalignments, all collected data were field-symmetrized and antisymmetrized.

### Calculations
First-principles calculations were performed based on the density functional theory implemented in the Vienna ab initio Simulation Package described by the projector augmented wave method[42–45]. The exchange-correlation interaction was included via generalized gradient approximation and parameterized using the Perbew–Burke–Ernzerhof functional[46]. The kinetic energy cut-off was set to 480 eV. A $13 \times 13 \times 6$ $k$-mesh was adopted for Brillouin zone sampling. The energy convergence criterion was set to $10^{-6}$ eV. Lattice parameters with $a = 4.963$ Å and $c = 7.752$ Å were used in the calculation. To calculate the Fermi surface, a Wannier tight-binding Hamiltonian based on the Ta 5$d$ and As 4$p$ orbitals was constructed using the Wannier90 package[47]. In order to determine the quantum oscillation frequencies, the Fermi pockets are examined by slicing them perpendicular to the magnetic field. The frequencies are then calculated from the local extrema using the Onsager relation. By comparing these calculated oscillation frequencies with experimental data, the Fermi level is determined. This is accomplished by testing various Fermi energies, as shown in Supplementary Fig. 1.

## Data availability
All data necessary to understand and assess this manuscript are shown in the main text and the Supporting Information. The data that support the findings of this study are available from the corresponding author upon reasonable request.

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

## Acknowledgements

This work was financially supported by the Deutsche Forschungsgemeinschaft (DFG) under SFB1143 (Project No. 247310070) and the Wuerzburg-Dresden Cluster of Excellence on Complexity and Topology in Quantum Matter ct.qmat (EXC 2147, Project No. 390858490). Y.P. acknowledges the financial support from the National Natural Science Foundation of China (Grant No. 52401263). The authors acknowledge Drs. Jingchen Yu and Yilin Jiang from Tsinghua University for their helpful discussions and assistance. Open access funding enabled and organized by Projekt DEAL.

## Author contributions

H.H., Y.P. and B.H. designed the work. X. F. carried out the DFT calculations. V.H. grew the single crystals. H.H. and B.H. carried out the crystallinity, composition characterization, and transport property measurements. H.W. helped with the measurements and analysis of the transport properties. C.F. supervised the project. All authors analyzed the results and co-edited the manuscript.

## Funding

## Competing interests

The authors declare no competing interests.
