## [Transparent Peer Review file · Nature Communications]

Multipocket synergy towards high thermoelectric performance in topological semimetal TaAs₂

Corresponding Author: Dr Bin He

Version 0:

Reviewer comments:

Reviewer #1

(Remarks to the Author)

In this paper, the authors have presented an experimental investigation of the thermoelectric effects in topological semimetal TaAs₂. They have demonstrated that the simultaneous presence of the compensated electron-hole bands and uncompensated gapped Dirac-like band leads to a simultaneous large Nernst signal and an unsaturated magneto-Seebeck coefficient. They have claimed that this multi-pocket synergy effect is presented here for the first time. However, I recognized that the same situation happens for NbP Weyl semimetal, and it has been reported in Ref. [36]. Nevertheless, the reported values for both longitudinal and transverse thermopowers are less than the present paper in the approximately same magnetic field which can be due to the polycrystalline samples used in that paper. Another concern is about the calculation of the electron and hole pockets. The authors considered that there was a large electron pocket around the Y point. However, according to the supplementary Fig. 1-a and the literature, a small hole pocket may be present at this point (see Ref. [27], Sci. Rep. 6, 27294 (2016), and Sabin Regmi et al 2024 J. Phys.: Condens. Matter 36 075502). Therefore, I have the following comments to the authors.

- 1) Authors of the paper should describe the novelty of their work in comparison to Ref. [36].
- 2) If the hole pocket is present around the Y point as reported by others, authors have to update their calculations. If this point is true, the electron and hole pockets may not compensate each other.
- 3) It is not clear how the phonon-drag contribution in the Nernst effect has been measured. The measured values of the Nernst signal in almost all temperatures are greater than the summation of the phonon-drag and carrier diffusion contributions. What can be the other contributions to the Nernst signal? This case is also present in the Seebeck effect and measured values are greater than the theoretically calculated Seebeck coefficients.
- 4) Why are the plots for density and mobility of the carriers in Fig. 2-c, d not presented at room temperature?
- 5) As shown in Fig. 3-f, this material possesses a considerable Seebeck coefficient (about -40) at room temperature and zero magnetic field. What can be the physical origin of this large value of thermopower in comparison to similar topological semimetals?
- 6) The figure of merit has not been plotted for the Seebeck coefficient.

Reviewer #2

(Remarks to the Author)

This paper discusses the thermoelectric properties of topological materials, addressing multiple transport states and illustrating their effects. The results are exciting; the authors observe high transverse thermoelectric transport. However, the manuscript lacks some physical mechanisms to support these observations. Additional evidence is needed, and the work requires revision.

The methodology is standard and meets the expected criteria, but more details should be provided and addressed. Before the final decision, the authors should consider the following issues:

- In the transport model, there are four different states: electrons and holes in the Dirac states, and electrons and holes in the nodal lines. The electrical transport should reflect a combination of these four states. The authors should calculate the Fermi level position based on the R_{xx} and R_{xy} measurements. Additionally, they need to further verify this through the combination of Shubnikov-de Haas (SdH) oscillation and density functional theory (DFT) calculations. A comparison between the SdH frequency and DFT calculations would be beneficial, as the intersection area between the Fermi surface and the magnetic field determines the oscillation frequency.
- The thermoelectric transport mechanism is somewhat confusing. The authors do not distinguish the effects of the four different states on thermoelectric transport, which can be indicated by their corresponding Fermi levels. Can the authors fit S_{xx} and S_{xy} together and compare the fitting parameters D and D_h with those from electrical transport?
- The authors should calculate the Seebeck effect, as this would provide insights into the different transport states.
- The enhanced Nernst effect may be attributed to the large Dirac cone. Berry curvature can lead to a larger transverse anomalous Nernst effect, which the authors did not discuss. Please see the attached references:
 - o Adv. Mater. 2024, 2311644
 - o Phys. Rev. Lett. 2017, 118, 136601
- Phonon drag requires strong electron-phonon coupling and large thermal conductivity. At what temperature did the authors conduct the phonon-drag calculations? Given that the thermal conductivity seems to be low at low temperatures, it is questionable whether phonon drag can have such a significant effect in this case.
- The thermoelectric oscillation frequency should match the SdH oscillation frequency. Did the authors observe any other oscillation frequencies?

Version 1:

Reviewer comments:

Reviewer #1

(Remarks to the Author)

The authors have satisfactorily considered all of the comments, so I can recommend this paper for publication in the NCOMMS.

November 12, 2024

Dear Editor and Reviewers,

Many thanks for your letter dated on Nov. 6, 2024, and the reviewer reports on our manuscript entitled “**Multipocket synergy towards high thermoelectric performance in topological semimetal TaAs₂**” (Manuscript number: NCOMMS-24-60378). We really appreciate the valuable comments from the referees and the chance you gave us for the revision. We hope the responses and revised manuscript address the concerns of the reviewers and meet the requirements of the esteemed journal *Nature Communications*. Our point-to-point responses to the reports of reviewers are listed as below:

Reviewer: 1

In this paper, the authors have presented an experimental investigation of the thermoelectric effects in topological semimetal TaAs₂. They have demonstrated that the simultaneous presence of the compensated electron-hole bands and uncompensated gapped Dirac-like band leads to a simultaneous large Nernst signal and an unsaturated magneto-Seebeck coefficient. They have claimed that this multi-pocket synergy effect is presented here for the first time. However, I recognized that the same situation happens for NbP Weyl semimetal, and it has been reported in Ref. [36]. Nevertheless, the reported values for both longitudinal and transverse thermopowers are less than the present paper in the approximately same magnetic field which can be due to the polycrystalline samples used in that paper. Another concern is about the calculation of the electron and hole pockets. The authors considered that there was a large electron pocket around the Y point. However, according to the supplementary Fig. 1-a and the literature, a small hole pocket may be present at this point (see Ref. [27], Sci. Rep. 6, 27294 (2016), and Sabin Regmi et al 2024 J. Phys.: Condens. Matter 36 075502). Therefore, I have the following comments to the authors.

Response:

We appreciate your valuable feedback. Regarding the concerns about the novelty and band calculations in our work, we would like to provide some clarifications that we believe will be beneficial.

Novelty: In 2018, Fu et al. reported the Nernst power factor in polycrystalline NbP, but the motivation and findings of our work differ significantly from theirs. According to their report, they utilized a combination of small and large electron pockets to compensate for a large hole pocket in order to achieve high Nernst thermopowers and power factors. However, they did not explore the role of the small pocket in the Seebeck coefficient, and the longitudinal power factors remained low across the entire temperature range. However, the exceptional compensated large hole pocket and large

electron pocket played a significant role in the high Nernst thermopowers and Nernst power factors observed in our study. Moreover, the presence of a third small Dirac pocket led to the remarkable magneto-Seebeck coefficients and longitudinal power factors due to the extreme quantum limit. As a result, we were able to achieve dual-high transverse and longitudinal thermopowers and power factors in TaAs₂.

Band calculations: We apologize for the mistake in the description provided. After carefully reviewing Supplementary Fig. 1a and the related literature [R1-R3], it has been clarified that there is only a small hole pocket around the Y point. Our actual intention in the manuscript was to highlight the existence of the large electron pocket around the M point. The manuscript has been corrected accordingly, and a thorough revision can be found in our response to Comment 1.2.

Followings are the point-by-point response, we hope it addresses your concerns.

Reference:

[R1] Wadge, A. S. *et al.* Electronic properties of TaAs₂ topological semimetal investigated by transport and ARPES. *J. Phys.: Condens. Matter* **34**, 125601 (2022).

[R2] Luo, Y. K. *et al.* Anomalous electronic structure and magnetoresistance in TaAs₂. *Sci. Rep.* **6**, 27294 (2016).

[R3] Regmi, S. *et al.* Electronic structure in a transition metal dipnictide TaAs₂. *J. Phys.: Condens. Matter* **36**, 075502 (2024).

Comment 1.1

Authors of the paper should describe the novelty of their work in comparison to Ref. [36].

Response 1.1:

Thanks for your insightful comment. In 2018, Fu et al. reported the Nernst power factor in polycrystalline NbP, but the motivation and findings of our work differ significantly from theirs. According to their report, they utilized a combination of small and large electron pockets to compensate for a large hole pocket in order to achieve high Nernst thermopowers and power factors. However, they did not explore the role of the small pocket in the Seebeck coefficient, and the longitudinal power factors remained low across the entire temperature range. However, the exceptional compensated large hole pocket and large electron pocket played a significant role in the high Nernst thermopowers and Nernst power factors observed in our study. Moreover, the presence of a third small Dirac pocket led to the remarkable magneto-Seebeck coefficients and longitudinal power factors due to the extreme quantum limit. As a result, we were able to achieve dual-high transverse and longitudinal thermopowers and power factors in TaAs₂. We have included relevant discussion in the manuscript.

Text changes in the manuscript:

Prior to this study, Fu et al. had already investigated the Nernst power factor in polycrystalline NbP through a multipocket strategy⁴⁰. Nevertheless, the motivation and outcomes of our study diverge significantly from theirs. While they relied on a combination of small and large electron pockets to compensate for a large hole pocket for enhanced Nernst thermopowers and power factors, they did not delve into the significance of the small pocket in the Seebeck coefficient.

Comment 1.2

If the hole pocket is present around the Y point as reported by others, authors have to update their calculations. If this point is true, the electron and hole pockets may not compensate each other.

Response 1.2: We apologize for the mistake in the description provided. After carefully reviewing Supplementary Fig. 1a and the related literature [R1-R3], it has been clarified that there is only a small hole pocket around the Y point. Our actual intention in the manuscript was to highlight the existence of the large electron pocket around the M point. The manuscript has been corrected accordingly.

Text changes in the manuscript:

The hole pocket is located at the M point, while the large electron pocket can be found close to the M point, and there is also a small electron pocket near point A.

Reference:

[R1] Wadge, A. S. *et al.* Electronic properties of TaAs₂ topological semimetal investigated by transport and ARPES. *J. Phys.: Condens. Matter* **34**, 125601 (2022).

[R2] Luo, Y. K. *et al.* Anomalous electronic structure and magnetoresistance in TaAs₂. *Sci. Rep.* **6**, 27294 (2016).

[R3] Regmi, S. *et al.* Electronic structure in a transition metal dipnictide TaAs₂. *J. Phys.: Condens. Matter* **36**, 075502 (2024).

Comment 1.3

It is not clear how the phonon-drag contribution in the Nernst effect has been measured. The measured values of the Nernst signal in almost all temperatures are greater than the summation of the phonon-drag and carrier diffusion contributions. What can be the other contributions to the Nernst signal? This case is also present in the Seebeck effect and measured values are greater than the theoretically calculated Seebeck coefficients.

Response 1.3:

Thank you for emphasizing this crucial point. In thermoelectrics, as temperature rises,

higher momentum phonons are activated. When the momentum of long-wave acoustic phonons aligns with that of carriers on the Fermi surface, the phonon-drag effect occurs. It is worth noting that the phonon-drag contribution in the Nernst effect cannot be measured or quantified through DFT calculations.

We apologize for the misunderstanding regarding Fig. 3b. Our calculations only considered the Nernst thermopowers associated with the charge carrier diffusion processes, as outlined in Supplementary Note I. Upon closer examination, we observed that the experimental Nernst thermopower (red line) reached a maximum value of $156 \mu\text{V K}^{-1}$ at 4.8 K and 9 T, which is significantly higher than the thermopower related to charge carrier diffusion processes (gray line, $\sim 2 \mu\text{V K}^{-1}$) at the same temperature and magnetic field. This discrepancy between the experimental and theoretical Nernst thermopowers (related to charge carrier diffusion processes) can be attributed to the phonon-drag effect. Thus, the assertion that the Nernst signal measurements consistently surpass the combined impacts of phonon-drag and carrier diffusion contributions at almost all temperatures is not sustainable.

As the temperatures differ between the theoretical and experimental results, direct comparison of the Seebeck coefficient values is not feasible. However, if we consider the values at 5 K as an example, it is evident that the theoretical values ($-45 \mu\text{V K}^{-1}$) are higher than the experimental value ($-38 \mu\text{V K}^{-1}$), not the other way around. Actually, the theoretical values only considered the small Dirac pocket in ideal conditions, overlooking the influence of the other two larger pockets. Therefore, the theoretical values in Fig. 3d may exhibit slight discrepancies from the experimental values in Fig. 3f.

Comment 1.4

Why are the plots for density and mobility of the carriers in Fig. 2-c, d not presented at room temperature?

Response 1.4:

Thanks for the careful question. We have plotted the carriers' concentration and mobility at temperatures ranging from 2 K to 100 K in Fig. 2c and d. However, we are unable to fit the actual electron and hole concentration and mobility at room temperature using the two-band model because the TaAs₂ crystal exhibits single-carrier behavior, as indicated by the Hall resistivity in Fig. R1. Moreover, since the mobility decreases rapidly with increasing temperature and the high Nernst thermoelectric performance shows up below 50 K, we focus on the low temperature range from 2 K to 50 K.

Figure R1 Hall resistivity ρ_{yx} as a function of the magnetic field at 300 K.

Text changes in the manuscript:

When the temperature ranges from 2–100 K, n_e and n_h exhibit a similar increase.

Figure changes in the manuscript:

Fig. 2 Electrical transport properties and quantum oscillations. c Carrier concentration. d Carrier mobility.

Comment 1.5

As shown in Fig. 3-f, this material possesses a considerable Seebeck coefficient (about -40) at room temperature and zero magnetic field. What can be the physical origin of this large value of thermopower in comparison to similar topological semimetals?

Response 1.5:

We appreciate the reviewer's insightful comment. As shown in Fig. R2, the Seebeck coefficient transitions from positive to negative as temperature increases at 0 T, reaching a peak value at 37.7 K due to the phonon drag effect. At temperatures above

37.7 K, electrons may start to have a notable impact on the Seebeck coefficient, potentially because of a slight change in the Fermi level induced by the high temperatures. This could result in an asymmetry between the conduction and valence bands [R4]. When this effect is combined with the influence of a small electron pocket, the Seebeck coefficient can reach $-43 \mu\text{V K}^{-1}$, consistent with previous research [R5, R6].

Figure R2 Temperature dependent thermal conductivity and Seebeck coefficient at 0 T of a TaAs₂ single crystal.

Reference:

[R4] Zhang, Y. *et al.* Electronic evidence of temperature-induced Lifshitz transition and topological nature in ZrTe₅. *Nat. Commun.* **8**, 15512 (2017).
[R5] Pan, Y. *et al.* Ultrahigh transverse thermoelectric power factor in flexible Weyl semimetal WTe₂. *Nat. Commun.* **13**, 3909 (2022).
[R6] Li, P. *et al.* Colossal Nernst power factor in topological semimetal NbSb₂. *Nat. Commun.* **13**, 7612 (2022).

Comment 1.6

The figure of merit has not been plotted for the Seebeck coefficient.

Response 1.6: Thanks for your constructive comment. The figure of merit has been plotted for the Seebeck coefficient and can be found in the Supplementary Fig. 8c-d.

Text changes in the manuscript:

Considering the large power factors and high thermal conductivity, we can obtain transverse (Z_N , $Z_N T$, defined as $Z_N = PF_N/\kappa$) and longitudinal figure of merits (Z_S , $Z_S T$, defined as $Z_S = PF_S/\kappa$, Supplementary Fig. 8); their variation trend is similar to that of PF_N and PF_S .

Figure changes in the Supporting Information:

Supplementary Figure 8. Nernst figure of merit. Magnetic field dependence of (a) Z_N , (b) $Z_N T$, (c) Z_S , and (d) $Z_S T$ at different temperatures.

Reviewer: 2

This paper discusses the thermoelectric properties of topological materials, addressing multiple transport states and illustrating their effects. The results are exciting; the authors observe high transverse thermoelectric transport. However, the manuscript lacks some physical mechanisms to support these observations. Additional evidence is needed, and the work requires revision.

The methodology is standard and meets the expected criteria, but more details should be provided and addressed. Before the final decision, the authors should consider the following issues:

Response:

We appreciate your recognition of this work. Below is a detailed response to each point and we have revised the manuscript based on your constructive comments.

Comment 2.1

In the transport model, there are four different states: electrons and holes in the Dirac states, and electrons and holes in the nodal lines. The electrical transport should reflect a combination of these four states. The authors should calculate the Fermi level position based on the R_{xx} and R_{xy} measurements. Additionally, they need to further verify this through the combination of Shubnikov-de Haas (SdH) oscillation and density functional theory (DFT) calculations. A comparison between the SdH frequency and DFT calculations would be beneficial, as the intersection area between the Fermi surface and the magnetic field determines the oscillation frequency.

Response 2.1:

Thank you for raising this insightful question.

For the different states: In the absence of the spin-orbit coupling (SOC), nodal lines were formed near points A and M, causing an anti-crossing band. However, when the SOC effect is considered, the nodal lines gap out the anti-crossing feature, leading to the formation of a massive Dirac fermion that aligns with the zero-field massive Dirac dispersion relation. As shown in Fig. 1b and Supplementary Fig. 1, there are three Fermi pockets intersecting the Fermi level E_F . The large hole pocket centered at the M point and the large electron pocket near the M point can result in an electron-hole compensation behavior, leading to large Nernst thermopowers. Additionally, the massive Dirac pockets (the small electron pocket) around point A may contribute to a significant magneto-Seebeck effect due to quantum effects.

For the Fermi level calculation: In order to determine the quantum oscillation frequencies, the Fermi pockets are examined by slicing them perpendicular to the magnetic field. The frequencies are then calculated from the local extrema using the Onsager relation. By comparing these calculated oscillation frequencies with

experimental data, the Fermi level is determined. This is accomplished by testing various Fermi energies, as shown in Supplementary Fig. 1a. Nevertheless, two experimental frequencies were not displayed in Supplementary Fig. 1a, possibly due to the following reason: Oscillation frequencies are determined by identifying the extreme cross-sections of Fermi pockets sliced on a $101 \times 101 \times 101$ k -grid. Due to the expansive nature of the hole pocket in momentum space, there is a possibility of overlooking some frequencies. To accurately map the complete frequency spectrum, a more extensive Fermi pocket may be required. By aligning experimental data with our calculations, we have positioned the Fermi level at -45 meV, as shown by the dashed line in Fig. 1(b) and Supplementary Fig. 1b. This alignment is also in line with the carrier density when compared to previous studies [R1]. Hence, the Fermi level we calculated in our study is appropriate and aligns with the experimental findings.

Reference:

[R1] Wadge, A. S. *et al.* Electronic properties of TaAs₂ topological semimetal investigated by transport and ARPES. *J. Phys.: Condens. Matter* **34**, 125601 (2022).

Text changes in the Method:

In order to determine the quantum oscillation frequencies, the Fermi pockets are examined by slicing them perpendicular to the magnetic field. The frequencies are then calculated from the local extrema using the Onsager relation. By comparing these calculated oscillation frequencies with experimental data, the Fermi level is determined. This is accomplished by testing various Fermi energies, as shown in Supplementary Fig. 1.

Figure changes in the Supporting Information:

Supplementary Figure 1. DFT calculation. a. Calculated Fermi level dependent frequency.

Comment 2.2

The thermoelectric transport mechanism is somewhat confusing. The authors do not distinguish the effects of the four different states on thermoelectric transport, which can be indicated by their corresponding Fermi levels. Can the authors fit S_{xx} and S_{yx} together and compare the fitting parameters D and D_H with those from electrical transport?

Response 2.2:

We appreciate the reviewer for providing this insightful comment. As depicted in Fig. 1b and Supplementary Fig. 1, there are three Fermi pockets intersecting the Fermi level E_F . The large hole pocket is situated at the M point, while the large electron pocket is located near the M point, with a small electron pocket close to point A. The electron-hole compensation behavior of the large hole and electron pockets leads to a significant Nernst thermopower due to the phonon-drag effect. Additionally, the presence of a massive Dirac pocket may result in a substantial magneto-Seebeck effect due to quantum effects. These key findings have been presented in the manuscript, with relevant theoretical and experimental evidence included to reinforce them.

Figure R3 The fitted parameters D and D_H from S_{xx} as a function of temperature.

Based on your questions, we also attempted to fit the parameters D and D_H by using the simplified Mott relation as follows:

$$S_{xx}(B) = A \left(\frac{\sigma_{xx}^2}{\sigma_{xx}^2 + \sigma_{yx}^2} D + \frac{\sigma_{yx}^2}{\sigma_{xx}^2 + \sigma_{yx}^2} D_H \right)$$

$$S_{yx}(B) = A \frac{\sigma_{xx} \sigma_{yx}}{\sigma_{xx}^2 + \sigma_{yx}^2} (D_H - D)$$

where $A = \frac{\pi^2 k_B^2 T}{3e}$. The parameters $D = \partial \ln \sigma / \partial \zeta$ and $D_H = \partial \ln \sigma_{yx} / \partial \zeta$ are independent of the carrier mobility, and ζ is the chemical potential. In Fig. R3, we fitted the D and D_H using the magneto-Seebeck coefficient S_{xx} , but the values obtained were deemed

unreasonable and did not offer any understanding of the transport properties mechanism. Since our material, TaAs₂, is an anisotropic system with a space group of *C*2/m, the parameters *D* and *D_H*, which originate from the Mott relation suitable for isotropic systems like Pb_{1-x}Sn_xSe [R7-R9], are not applicable here. Additionally, the transport properties in TaAs₂ can be well explained by the phonon-drag effect and the quantum limit of the massive Dirac pocket, as discussed in Comment 2.3 and 2.4. Once again, we appreciate the reviewer's valuable comment.

Reference:

[R7] Ouyang, W. K. *et al.* Extraordinary Thermoelectric Properties of Topological Surface States in Quantum-Confined Cd₃As₂ Thin Films. *Adv. Mater.* **36**, 2311644 (2024).

[R8] Liang, T. *et al.* Anomalous Nernst Effect in the Dirac Semimetal Cd₃As₂. *Phys. Rev. Lett.* **118**, 136601 (2017).

[R9] Liang, T. *et al.* Evidence for massive bulk Dirac fermions in Pb_{1-x}Sn_xSe from Nernst and thermopower experiments. *Nat. Commun.* **4**, 2696 (2013).

Comment 2.3

The authors should calculate the Seebeck effect, as this would provide insights into the different transport states.

Response 2.3:

Thank you for your constructive suggestion. To provide insights into the impact of two compensated pockets on transport properties, we utilized a two-carrier model to calculate the Seebeck coefficient of carriers S_{xx}^i in the two compensated pockets (where *i* represents carrier types), Seebeck coefficient related to charge carrier diffusion processes S_d^i , and phonons S_p^i . We observed that S_{xx}^e and S_{xx}^h show non-linear temperature dependence below 100 K, with peaks around 45 K, indicating a significant contribution from phonons to the Seebeck coefficient at low temperatures. The absolute values of S_p^e and S_p^h peak at 73 $\mu\text{V K}^{-1}$ and 76 $\mu\text{V K}^{-1}$ around 45 K, respectively, which are much larger than the values of S_{de} (4.1 $\mu\text{V K}^{-1}$) and S_{dh} (2.6 $\mu\text{V K}^{-1}$) at the same temperature. This suggests that the phonon-drag effect significantly enhances the total Nernst effect in TaAs₂ single crystal.

Moreover, Fig. 3d illustrates the theoretical magneto-Seebeck effect of a massive Dirac pocket, providing further evidence that the significant Seebeck coefficient is a result of a massive Dirac fermion. We have included additional discussions on the Seebeck coefficient in relation to various states, and further details can be found in the manuscript and Supplementary Information.

Text changes in the manuscript:

Moreover, a two-carrier model was used to analyze the behavior of the Nernst thermopower in TaAs₂. The Seebeck coefficients for electrons (S_{xx}^e) and holes (S_{xx}^h) deviate from linear temperature dependence below 100 K and exhibit peaks around 45 K, indicating a significant contribution from phonons (S_p) to the Seebeck coefficient at low temperatures. The absolute values of S_p^e and S_p^h reach maximum values of 73 $\mu\text{V K}^{-1}$ and 76 $\mu\text{V K}^{-1}$ around 45 K, which are much larger than the values of S_d^e (4.1 $\mu\text{V K}^{-1}$) and S_d^h (2.6 $\mu\text{V K}^{-1}$) at the same temperature. This suggests that the phonon-drag effect significantly enhances the total Nernst effect in single-crystalline TaAs₂^{10,31}. Additional details can be found in Supplementary Note I and Supplementary Fig. 6. Additionally, the thermal conductivity behavior (Supplementary Fig. 7) can be used as further evidence to support the phonon-drag effect.

Text changes in the Supporting Information:

Supplementary Note I. Calculation of thermopowers

In a typical two-carrier model, when $\sigma_{yx}^2 \ll \sigma_{xx}^2$, the S_{xx} and S_{yx} can be written as:

$$S_{xx} = \frac{S_{xx}^e (\sigma_{xx}^e \sigma_{xx} + \sigma_{yx}^e \sigma_{xx}) + S_{xx}^h (\sigma_{xx}^h \sigma_{xx} + \sigma_{yx}^h \sigma_{xx})}{\sigma_{xx}^2}$$
$$S_{yx} = \frac{S_{xx}^e (\sigma_{yx}^e \sigma_{xx} - \sigma_{xx}^e \sigma_{yx}) + S_{xx}^h (\sigma_{yx}^h \sigma_{xx} - \sigma_{xx}^h \sigma_{yx})}{\sigma_{xx}^2}$$

For simplicity, the Nernst thermopower S_{yx} can be expressed as:

$$S_{yx} = \frac{\sigma_{xx}^e \sigma_{xx}^h (\mu_e + \mu_h) B}{(\sigma_{xx}^e + \sigma_{xx}^h)^2} (S_{xx}^h - S_{xx}^e)$$

where σ_{yx}^e , σ_{yx}^h , σ_{xx}^e , σ_{xx}^h , μ_e , μ_h , B , S_{xx}^h , and S_{xx}^e represent electron Hall conductivity, hole Hall conductivity, electron electrical conductivity, hole electrical conductivity, electron mobility, hole mobility, magnetic field, hole Seebeck coefficient and electron Seebeck coefficient, respectively.

In the degenerate limit, the Seebeck coefficient S_d related to the charge carrier diffusion processes can be expressed as:

$$S_d = \frac{8\pi^2 m^* k_B^2 T}{3eh^2} \left(\frac{\pi}{3n} \right)^{2/3}$$

where m^* , k_B , T , e , h , and n are effective mass, Boltzmann constant, temperature, elementary charge, Planck constant, and carrier concentration, respectively.

Hence, the Seebeck coefficient S_p related to the phonons can be written as:

$$S_p = S_{xx} - S_d$$

Figure changes in the Supporting Information:

Supplementary Figure 6. Thermal transport properties. **a** Respective Seebeck coefficient of electrons S_{xx}^e and holes S_{xx}^h under 5 T for TaAs₂ derived from the two-carrier model. **b** Seebeck coefficient of electrons and holes related to the charge carrier diffusion processes (S_d^e and S_d^h) and phonons (S_p^e and S_p^h) at 5 T, respectively. **c** Temperature dependence of the difference between Seebeck coefficient of electrons and holes ($S_{xx}^h - S_{xx}^e$) of TaAs₂ under 5 T. **d** Seebeck coefficient and Nernst thermopower oscillations of TaAs₂ after subtracting the background as a function of $1/B$.

Comment 2.4

The enhanced Nernst effect may be attributed to the large Dirac cone. Berry curvature can lead to a larger transverse anomalous Nernst effect, which the authors did not discuss. Please see the attached references:

o Adv. Mater. 2024, 2311644

o Phys. Rev. Lett. 2017, 118, 136601

Response 2.4:

Thank you very much for your valuable suggestion. The topological semimetal TaAs₂ has a centrosymmetric monoclinic structure with a space group of $C2/m$ (no. 12). Therefore, the Berry curvature is zero and the anomalous Nernst effect is not considered in this material, even under magnetic since the external magnetic field is insufficient to

induce a decent non-zero Berry curvature in TaAs₂, which is different from Cd₃As₂ observed by previous studies [R10, R11]. We have added related discussion and references in the manuscript.

Reference:

[R10] Ouyang, W. K. *et al.* Extraordinary Thermoelectric Properties of Topological Surface States in Quantum-Confined Cd₃As₂ Thin Films. *Adv. Mater.* **36**, 2311644 (2024).

[R11] Liang, T. *et al.* Anomalous Nernst Effect in the Dirac Semimetal Cd₃As₂. *Phys. Rev. Lett.* **118**, 136601 (2017).

Text changes in the manuscript:

Since the topological semimetal TaAs₂ has a centrosymmetric monoclinic structure with a space group of C2/m (no. 12), the Berry curvature is zero and the anomalous Nernst effect is not considered, even under magnetic since the external magnetic field is insufficient to induce a decent non-zero Berry curvature in TaAs₂, which is different from Cd₃As₂ observed by previous studies^{21,22}.

Reference changes in the manuscript:

21 Ouyang, W. K. *et al.* Extraordinary Thermoelectric Properties of Topological Surface States in Quantum-Confined Cd₃As₂ Thin Films. *Adv. Mater.* **36**, 2311644 (2024).

22 Liang, T. *et al.* Anomalous Nernst Effect in the Dirac Semimetal Cd₃As₂. *Phys. Rev. Lett.* **118**, 136601 (2017).

Comment 2.5

Phonon drag requires strong electron-phonon coupling and large thermal conductivity. At what temperature did the authors conduct the phonon-drag calculations? Given that the thermal conductivity seems to be low at low temperatures, it is questionable whether phonon drag can have such a significant effect in this case.

Response 2.5:

Thanks for your insightful comment and we agree with the referee that phonon-drag effect requires strong electron-phonon coupling and large thermal conductivity. To provide a clearer insight into the effect of phonon drag on the Nernst thermopower, we utilized a two-carrier model to calculate the Seebeck coefficient of carriers S_{xx}^i (where i represents carrier types) in the two compensated pockets, Seebeck coefficient related to charge carrier diffusion processes S_d^i , and phonons S_p^i . We observed that S_{xx}^e and S_{xx}^h show non-linear temperature dependence below 100 K, with peaks around 45 K,

indicating a significant contribution from phonons to the Seebeck coefficient at low temperatures. Even at around 8 K, the absolute values of S_p^e and S_p^h can reach $21.4 \mu\text{V K}^{-1}$ and $8.3 \mu\text{V K}^{-1}$ around 45 K, which are much larger than the values of S_d^e ($0.8 \mu\text{V K}^{-1}$) and S_d^h ($0.5 \mu\text{V K}^{-1}$).

Moreover, as shown in Fig. R2, TaAs₂ displays clear peak values for both thermal conductivity and Seebeck coefficient at the same temperature due to phonon-drag effect [R5, R6]. Notably, TaAs₂ exhibits a high thermal conductivity of approximately $62 \text{ W m}^{-1} \text{ K}^{-1}$ at 8 K, significantly surpassing the thermal conductivity of NbSb₂ ($\sim 6 \text{ W m}^{-1} \text{ K}^{-1}$) which still displays a substantial phonon drag effect at the same temperature [R6]. These discussions suggest that the phonon-drag effect significantly enhances the Nernst effect in single-crystalline TaAs₂ even at low temperature.

Reference:

[R5] Pan, Y. *et al.* Ultrahigh transverse thermoelectric power factor in flexible Weyl semimetal WTe₂. *Nat. Commun.* **13**, 3909 (2022).

[R6] Li, P. *et al.* Colossal Nernst power factor in topological semimetal NbSb₂. *Nat. Commun.* **13**, 7612 (2022).

Comment 2.6

The thermoelectric oscillation frequency should match the SdH oscillation frequency. Did the authors observe any other oscillation frequencies?

Response 2.6:

Thank you for your insightful question. Based on the Shubnikov–de Haas (SdH) oscillations results (shown in Figure R4), it is evident that there are five fundamental frequencies at $F = 11 \text{ T}$ (α), 45 T (β_1), 79 T (β_2), 179 T (β_3), and 373 T (γ), which align well with previous findings. Quantum oscillations are also clearly observed in both the Nernst thermopower and Seebeck coefficient below 8.3 K. Further analysis using Fast Fourier Transform (FFT) of the S_{xx} and S_{yx} oscillations at 4.8 K reveals six main frequencies, consistent with the SdH results. Additionally, an additional frequency β_4 (106 T) is observed in thermal transport oscillation, possibly related to an irregular hole pocket as reported previously [R1, R12]. The thermal transport oscillation, being the energy derivative, is more sensitive than the SdH, explaining why this frequency is not observed in the resistivity SdH analysis [R12]. The manuscript has been corrected accordingly, and a thorough revision can be found below.

Figure R4 Fast Fourier transform (FFT) spectra of magnetoresistivity, S_{xx} and S_{yx} at different temperatures.

Reference:

[R1] Wadge, A. S. *et al.* Electronic properties of TaAs₂ topological semimetal investigated by transport and ARPES. *J. Phys.: Condens. Matter* **34**, 125601 (2022).

[R12] Alam, M. S. *et al.* Temperature-driven spin-zero effect in TaAs₂. *J. Phys. Chem. Solids* **170**, 110939 (2022).

Text changes in the manuscript:

Moreover, an additional frequency β_4 (106 T) is observed, possibly related to an irregular hole pocket as reported previously. The thermal transport oscillation, being the energy derivative, is more sensitive than the SdH, explaining why this frequency is not observed in the SdH analysis^{29,34}.

Figure changes in the manuscript:

Fig. 3 Thermoelectric thermopowers. The inset shows the FFT spectra of S_{xx} and S_{yx} at 4.8 K.

We greatly appreciate the reviewers for evaluating our paper and providing constructive comments and suggestions. Your feedback has been invaluable in improving our manuscript and guiding our future research. We hope you and the reviewers will be satisfied with the revised version.

Yours sincerely,

Claudia Felser

December 5, 2024

Dear Reviewers,

Many thanks for the valuable reports on our manuscript entitled “**Multipocket synergy towards high thermoelectric performance in topological semimetal TaAs₂**” (Manuscript number: NCOMMS-24-60378A). We really appreciate the professional comments from the referees. Our point-to-point responses to the reports are listed as below:

Reviewer: 1

The authors have satisfactorily considered all of the comments, so I can recommend this paper for publication in the NCOMMS.

Response:

We are grateful for your recognition. Thank you very much for your efforts.

We greatly appreciate the reviewers for evaluating our paper and providing constructive comments and suggestions. Your feedback has been invaluable in improving our manuscript and guiding our future research.

Yours sincerely,

Claudia Felser
